

# Accurate 3D radiative transfer simulation of spectral solar irradiance during the total solar eclipse of August 21, 2017

Paul Ockenfuß[1], Claudia Emde[1], Bernhard Mayer[1], and Germar Bernhard[2]

[1]Meteorological Institute, Ludwig-Maximilians-University, Theresienstr. 37, 80333 Munich, Germany
[2]Biospherical Instruments Inc., San Diego, CA 92110, USA

**Correspondence:** Paul Ockenfuß (paul.ockenfuss@physik.uni-muenchen.de)

**Abstract.** We calculate the variation of spectral solar irradiance in the umbral shadow of the total solar eclipse of August 21, 2017 and compare it to observations. Starting from the sun's and moon's positions, we derive a realistic profile of the lunar shadow at the top of the atmosphere, including the effect of solar limb darkening. Subsequently, the Monte-Carlo model MYSTIC is used to simulate the transfer of solar radiation through the earth's atmosphere. Among the effects taken into account

are the atmospheric state (pressure, temperature), concentrations of major gas constituents and the curvature of the earth, as well as the reflectance and elevation of the surrounding area. We apply the model to the total solar eclipse on August 21, 2017 at a position located in Oregon, where irradiance observations were performed for wavelengths between 306 nm and 1020 nm. The influence of the surface reflectance, the ozone profile and mountains surrounding the observer is investigated. An increased sensitivity during totality is found for the reflectance and topography, compared to non-eclipse conditions. During the eclipse,

the irradiance at the surface does not only depend on the total ozone column (TOC) but also the vertical ozone distribution, which in general complicates derivations of the TOC from spectral surface irradiance. The findings are related to an analysis of the prevailing photon path and its difference compared to non-eclipse conditions. Using the most realistic estimate for each parameter, the model is compared to the irradiance observations. During totality, the relative difference between model and observations is less than 10% in the spectral range from 400 nm to 1020 nm. Slightly larger deviations occur in the ultraviolet

range below 400 nm and at 665 nm.

## 1 Introduction

When modeling real world processes, we generally want to achieve one of two main purposes: if measurements have not yet been performed, we can use the simulation to make predictions about the expected outcome and give recommendations for

the experimental setup. If however, measured data exists, we can validate and enhance our conceptual understanding of the phenomenon under investigation by comparison with the model results. In this case, one can also use the measurements to



verify the model itself, in order to increase the credibility in model predictions. The following study is carried out with the intention to serve all of these points.

In the past, the influence of a solar eclipse on surface radiation measurements and the derived state of the atmosphere were discussed in several studies. Zerefos et al. (2000) performed measurements in the UV domain during the total solar eclipse 1999 over Europe. They discussed the implications of solar limb darkening and the ratio of diffuse to direct irradiance during a solar eclipse on ozone measurements.

Kazantzidis et al. (2007) further examined the temporal change in the wavelength ratios for ultraviolet and visible wavelengths. Along with this study, the radiative transfer model MYSTIC (Mayer (2009); Emde and Mayer (2007)) was used to model the observations in three dimensions for the first time. The wavelengths discussed were 312 nm, 340 nm and 380 nm. During totality, the 3D model reproduced the measured irradiance at 380 nm with up to 5% accuracy, but deviated from the measurements by a factor of three for 312 nm. Kazantzidis et al. (2007) already pointed out that the ozone profile as well as the surrounding topography could possibly have an influence on the surface irradiance, which was not yet included in their model. The detailed model functionality is described in Emde and Mayer (2007), where the usefulness of a total solar eclipse to validate 3D radiative transfer codes was pointed out as well: in contrast to e.g. large scale cloud fields, the shape of the lunar shadow exhibits a very well defined geometry and therefore allows for an accurate representation of the reality within the model. Moreover, a 3D model is essential to simulate irradiance in the umbral shadow, where every detected photon has to be scattered from regions outside the shadow towards the observer.

For the total solar eclipse over North America on August 21, 2017, different groups performed irradiance measurements, e.g. Calamas et al. (2018) and Bernhard and Petkov (2019). The latter measured solar irradiance at 14 different wavelengths between 306nm and 1020nm with a sensitivity high enough to yield significant results also during totality. In their publication, they discussed the time outside the totality, i.e. the pre-umbra region, extensively, concluding that changes in the ozone column derived from spectral irradiance measurements reported in previous studies could potentially be attributed to uncertainties in the limb darkening parametrization. The importance of limb darkening, in particular when deriving ozone, is also supported by findings from Groebner et al. (2017) for the solar eclipse on March 20, 2015. To get a general overview of the results from many studies on atmospheric changes caused by solar eclipses, the reader is referred to Aplin et al. (2016)

In this study, we perform 3D model calculations for the conditions during the observations at totality by Bernhard and Petkov (2019). The comparison of the modelling results to the observations is an validation of our 3D radiative transfer model from ultraviolet to near infrared. Moreover, we continue the work of Emde and Mayer (2007) and Kazantzidis et al. (2007) by analyzing the model's sensitivity towards changes in the ozone column and profile, as well as the surface reflectance and topography, in order to understand how the environmental conditions influence the radiative fluxes during totality. Altogether, this provides us with the ability to precisely estimate the time evolution of radiative fluxes for the eclipse analyzed here as well as for upcoming eclipses. Where this gets important is during the totality, when the sun is completely covered by the lunar disk. Because intensities are reduced by up to four orders of magnitude compared to the uncovered sun, common measurement devices often operate close to their detection limit. Astronomical devices could produce better results at this time, but due to





their high sensitivities, they would have to be protected as soon as intensities start to increase again, which can be a matter of only a few minutes or seconds. Therefore, pre-estimating irradiance in advance is crucial to prepare measurement campaigns.

## 2 Methods and Techniques

The basis for modeling irradiances during totality of the solar eclipse is the Monte Carlo solver MYSTIC (Monte Carlo code
for the phYSically correct Tracing of photons In Cloudy atmospheres) (Mayer (2009)) with the adaptions already made in a previous study (Emde and Mayer, 2007). MYSTIC is operated as one of several radiative transfer solvers of the libRadtran package (www.libradtran.org, Mayer and Kylling (2005), Emde et al. (2016)) We enhanced and generalized this model in different aspects to make calculations at arbitrary observer positions on earth possible. In general, the simulation process can be divided into two distinct parts: in the celestial part, we calculate the distribution of solar irradiance under eclipse conditions in
a plane tangential to the top of atmosphere (TOA) for given positions of sun and moon. We call this plane the "sampling plane" $SP$, as illustrated in Figure 1. In the atmospheric part, the Monte Carlo model is used to determine the relative influence of every pixel in the sampling plane to the measured result at the observer position under non-eclipse conditions. This function is called the "contribution function" $C$. Multiplication of the contribution function with the irradiance distribution and subsequent summation yields the final model result. In this approach, the atmospheric part, which is costly to simulate, depends only on
the solar position. Therefore, it is sufficient to calculate this part only once for the total eclipse with a duration typically around two minutes. The fast movement of the lunar shadow is contained in the easier to calculate celestial part.

### 2.1 Celestial part

Ephemeris data of sun and moon was gathered using Giorgini et al. (1996). The service provides access to the JPL Horizons system operated by the Solar System and Dynamics Group of the Jet Propulsion Laboratory (JPL). It offers the possibility to
create ephemeris for various celestial bodies in the solar system for arbitrary times and observer positions on earth. The data we used were azimuth, elevation, angle diameter and distance from observer respectively for sun and moon. These quantities can be interpreted as spherical coordinates of sun and moon in a local coordinate system centered at the observer. The calculation of extraterrestrial solar irradiance for a given position, time and wavelength $\lambda$ was done following the geometrical considerations in Koepke et al. (2001). The basic idea is to integrate the solar irradiance over the visible part of the solar disk, i.e. the part
which is not covered by the lunar disk. The key quantities in the calculation of the relative solar irradiance $w(X, R_M, \lambda)$ are the celestial distance $X$ of solar and lunar disk and the radius of the lunar disk $R_M$. For this and all further calculations, celestial distances are always measured in radii of the solar disk $R_S$, where $R_S$ is the angle radius of the solar disk in *arcsec*. An overview over the definition of the different geometrical quantities we use is given in Figure 2. In order to get the celestial distance $X(\boldsymbol{d}, t)$ at arbitrary positions $\boldsymbol{d}$ relative to the observer, we have to calculate the angle between the position vectors of
Sun and Moon as seen from $\boldsymbol{d}$. We obtain:

$$X(\boldsymbol{d}, t) = \arccos \left( \frac{(\boldsymbol{P_S} - \boldsymbol{d}) \cdot (\boldsymbol{P_M} - \boldsymbol{d})}{|\boldsymbol{P_S} - \boldsymbol{d}| \, |\boldsymbol{P_M} - \boldsymbol{d}|} \right) \tag{1}$$





with $\boldsymbol{P_S}(\boldsymbol{t})$ and $\boldsymbol{P_M}(\boldsymbol{t})$ the position vectors of sun and moon relative to the observer at time $t$. While the information about the shape of the lunar shadow is mainly contained in the spatial variation of $X$, the variation of $R_M(\boldsymbol{d},t)$ provides a small correction. An expression can be derived from the geometry shown in Figure 2 with the distance $|\boldsymbol{P_M} - \boldsymbol{d}|$ of the moon and the actual lunar radius which was set to $Rad_M = 1737.4$ km (Archinal et al. (2018)).

$$R_M(\boldsymbol{d},t) = \arcsin\left(\frac{Rad_M}{|\boldsymbol{P_M} - \boldsymbol{d}|}\right) \tag{2}$$

Evaluating Equation 2 directly at the observer position ($\boldsymbol{d} = 0$) shows a maximum deviation of 0.02 arcsec from the angle diameter predicted directly by JPL Horizons. Evaluations in 500 km distance of the observer change the lunar radius by $\pm 1$ arcsec. For the sun, a similar calculation shows that the relative change of the disk radius due to movements in the sampling plane is of the order $10^{-6}$ because of the much greater distance to the earth, so it is safe to set $R_S(\boldsymbol{d},t) = R_S(t)$. Because every evaluation of $w(X, R_M, \lambda)$ requires an expensive numerical integration over the visible solar disk, in the first step we precalculate $w$ for 4000 values of $X$ and nine values of $R_M$. In the second step, we calculate $X(\boldsymbol{d},t)$ and $R_M(\boldsymbol{d},t)$ for every pixel in the sampling plane and get $w$ by linear interpolation. In Figure 3, $w(X, R_M, \lambda)$ is shown for $\lambda = 555$ nm at time 17:20.

An important fact one has to consider when integrating over the visible part of the solar disk is the darkening towards the outside of the disk. The cause is that the sun has a diffuse border with an optical thickness like an atmosphere. Looking directly at the center, we can see to deeper and therefore hotter layers than looking at the rim. The parametrization for this "limb darkening" effect used by Koepke et al. (2001) is based on theoretical considerations from Waldmeier (1955). For our study, we follow the recommendations by Bernhard and Petkov (2019) and implemented the more accurate parametrization developed by Pierce et al. (1977), based on two measurement campaigns (Pierce and Slaughter (1977), Pierce et al. (1977)). The limb darkening factor $\Gamma$ describes the radiance relative to the center of the solar disk and is expressed as a fifth degree polynomial in $\mu = \cos(\Psi(r))$. Definition of $\Psi$ is shown in Figure 4. The coefficients are given for discrete wavelengths $\lambda_i$ in the spectral range from 304 nm to 1046 nm. In between, the values were interpolated linearly. The effect of limb darkening for different wavelengths is illustrated in Figure 5.

## 2.2 Atmospheric Part

To obtain the two dimensional contribution function at TOA, the fully spherical 3D Monte Carlo solver MYSTIC was applied (Mayer (2009); Emde and Mayer (2007)). For the details of this part of the simulation process, the reader is referred to Emde and Mayer (2007). In this setup, the photons are starting from the observer and traced backwards through the atmosphere until they hit the TOA or one of the domain boundaries. In the latter case, they are destroyed. This is necessary, because in spherical geometry, periodic boundary conditions are not applicable, unless the whole planet is simulated. We ensure that the boundaries have no implications on the result by use of a sufficiently large domain. To speed up the convergence, the "local estimate technique" (Marshak and Davis, 2005) is applied.



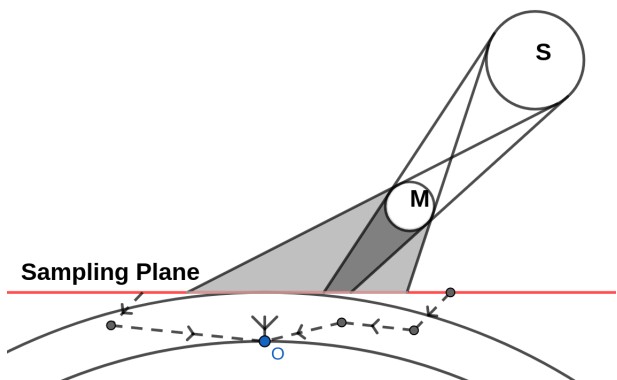

**Figure 1.** Schematic illustration of a solar eclipse, including umbral (dark gray) and penumbral (light gray) shadow and the definition of the sampling plane. $S$ denotes the sun, M the moon and $O$ the observer. The dashed lines show exemplary photon paths. In the atmospheric part, the solar angle is assumed to be constant, atmospheric refraction is neglected.

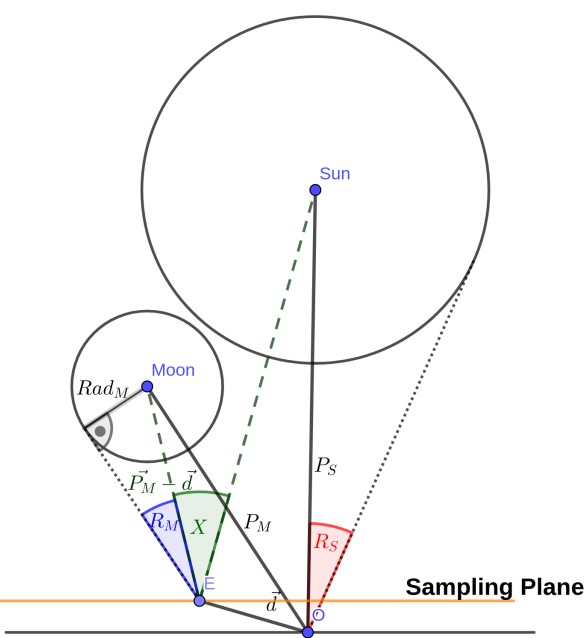

**Figure 2.** Graphical illustration (not to scale) of sun and moon with definitions of various geometrical quantities. $R_S$, $P_M$ and $P_S$ are obtained from JPL Horizons and allow calculation of $R_M$ and $X$ for arbitrary $\boldsymbol{d}$.

## 2.3 Modeled irradiance at location of observer

With the solar irradiance weighting function $w(\boldsymbol{d}, \lambda, t)$ and the contribution function $C(\boldsymbol{d}, \lambda, t)$ both available, we are now finally able to bring the parts together to obtain the irradiance at the sensor position. Summing up the product of $w$ and $C$ for





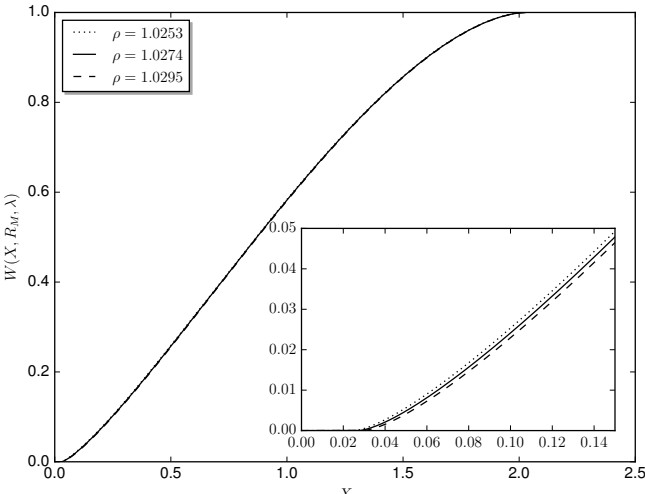

**Figure 3.** The relative solar irradiance $w(X, R_M, \lambda)$ compared to the uncovered sun, calculated for $\lambda = 555$ nm with the limb darkening parametrization from Pierce and Slaughter (1977) for different $\rho := R_M/R_S$.

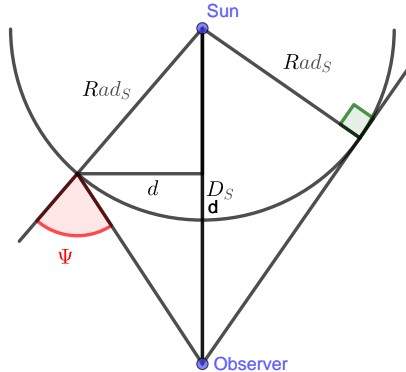

**Figure 4.** Definition of the angle $\Psi$ used in the equations for the limb darkening coefficient. If $d$, $D_S$ and $Rad_S = 695.700$ km are given, it is possible to derive $\Psi$ using simple trigonometric relations.

every point in the sampling plane yields the diffuse fraction of the sunlight reaching the sensor. Because the light of the sun reaching the sensor directly without any scattering in the atmosphere is not included in the inverse Monte Carlo results, we have to add it as an extra term $w(\boldsymbol{d_0}, \lambda, t) \cdot e^{-\tau}$. The exponential factor describes the direct atmospheric transmittance, depending on the total optical thickness $\tau$ which is obtained using libRadtran. $\boldsymbol{d_0}$ is the position where the line from the observer to the sun intersects the sampling plane. Multiplying the sum of direct and diffuse fraction of sunlight at the observer position with the extraterrestrial spectrum (ETS) under non-eclipse conditions, finally gives the values we were looking for. The factor $\kappa(t)$ corrects the ETS for variations in the earth-sun distance by use of the formula from Iqbal (1983). Because its time dependence





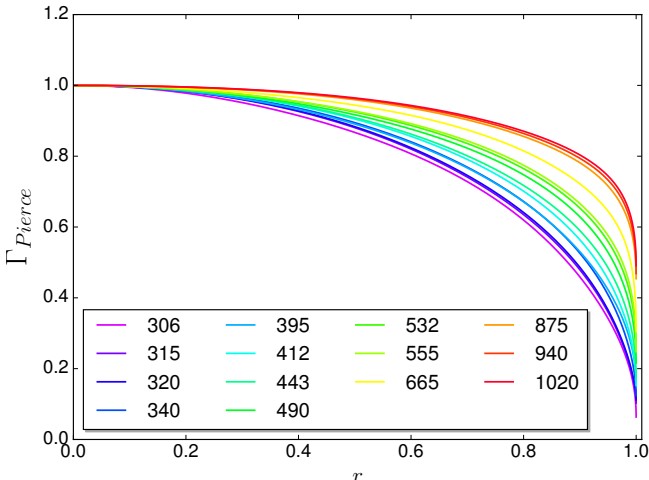

**Figure 5.** Values of the limb darkening factor $\Gamma$ from Pierce. Curves are for different wavelengths, given in nm. $r$ denotes the position on the visible solar disk, with $r = 0$ at the center and $r = 1$ at the limb.

is in the order of weeks, it is kept constant during the eclipse. The final expression reads:

$$I(\lambda,t) = \kappa \cdot ETS(\lambda) \cdot \left( e^{-\tau} \cdot w(\boldsymbol{d_0}, \lambda, t) + \sum_{\boldsymbol{d} \in SP} C(\boldsymbol{d}, \lambda, t) \cdot w(\boldsymbol{d}, \lambda, t) \right) \tag{3}$$

## 2.4 The eclipse 2017

In the following, we will apply the model to simulate solar irradiance for the total solar eclipse 2017 at a location in Smith
Rock State Park near Terrebonne, Oregon. Irradiance measurements from this site are available from Bernhard and Petkov
(2019). At first, we will rebuild the environmental and atmospheric conditions at the measurement site, starting from a basic
setup which will serve as a proof of concept. Stepwise, we will go on to more specific settings and analyze their sensitivity
towards the result. Finally, we will simulate time series of solar irradiance for the measured wavelengths and compare them to
the observed values.

The eclipse 2017 occurred on August 21 in the northern hemisphere, with the umbral shadow starting in the central pacific,
passing over the area of the United States and finally going further across two-thirds of the Atlantic Ocean. The penumbral
shadow hit the US western coastline at about 16:00UTC , before the umbral shadow has its first contact with the earth in
the pacific at 16:48UTC. In the following, all times will be given in UTC. Measurements were taken at $121°08'22.8''$W and
$44°21'46.6''$N. At this place, approximately 37 km away from the umbral shadow's center line, the totality started at 17:19:46
and lasted for 1:22 min. The detailed start and end times at the measurement site can be obtained from Table 1. The ratio of
lunar and solar disk radius varies for the time of the partial eclipse between 1.0250 and 1.0290. The diameters of the minor
axis of the elliptical shadow calculates to 3436 km for the penumbral shadow at the time of totality, and 94 km for the umbral
shadow.



The celestial data was created with JPL Horizons at the position $121°08'22.8''$W, $44°21'46.6''$N and an altitude of 866.8 m. From 14:30 until 20:00 values were taken every 120 s, while from 17:10 until 17:30, the time around the totality, a higher resolution of 5 s was chosen. Up to 675.5 nm the spectrum 2004JD004937-ETS_GUEYMARD described in Bernhard et al. (2004) and available at http://uv.biospherical.com/Version2/Paper/2004JD004937-ETS_GUEYMARD.txt was used. Above
675.5 nm, the extraterrestrial spectrum published by Gueymard (2004) was used. Both were corrected for day 233 of the year (August 21). Profiles of pressure, temperature, density and the major gas concentrations ($O_3$, $O_2$, $H_2O$, $CO_2$, $NO_2$) were taken from the midlatitude summer profiles by Anderson et al. (1986). The thereby specified molecular ozone profile was scaled to a total ozone column of 298 DU, corresponding to the value derived in Bernhard and Petkov (2019). The other profiles are not modified and aerosol is not included. Following the experiences made previously in Emde and Mayer (2007), a sampling grid
of 2000 km x 2000 km was used in the horizontal with 1 km step size for the contribution function at TOA. $10^7$ photons were traced for each wavelength.

| Event | Time[UT] | Elev[°] | Azim[°] |
|---|---|---|---|
| Start partial (1. contact) | 16:06:29 | 29.4 | 102.7 |
| Start total (2. contact) | 17:19:46 | 41.6 | 118.9 |
| Maximum | 17:20:27 | 41.7 | 119.1 |
| End total (3. contact) | 17:21:08 | 41.8 | 119.3 |
| End partial (4. contact) | 18:41:05 | 52.6 | 143.6 |

**Table 1.** Times and solar elevation, azimuth for different characteristic events of the eclipse 2017 at the measurement position. Elevation is given relative to the horizon, azimuth is measured clockwise from the north (JPL convention). Created with use of Espenak (2018).

## 3   Results

### 3.1   Time series

Figure 6 gives an overview over the change of irradiance with time between 10 min before and after totality. As described
above, the celestial part was fully time resolved, including the sun's position. The atmospheric part is based on one Monte Carlo simulation at 17:20. Comparing with Fig. 13 in Emde and Mayer (2007), we can see the same qualitative behavior. All wavelengths undergo a sudden drop as soon as totality occurs. The spectral irradiance of the 315 nm wavelength is remarkably lower than the visible irradiances, which is mainly due to ozone absorption. The strong increase of the Rayleigh scattering cross section towards shorter wavelengths is the reason for the stronger variation in irradiance during the totality period between
17:19 and 17:21 at 315 nm. Looking at the 875 nm near infrared wavelength, one finds that it decreases by nearly four orders of magnitude, leading to totality intensities almost as low as in the UV-part of the spectrum around 315 nm. The reason is again the proportionality of the Rayleigh scattering cross section to $\frac{1}{\lambda^4}$, making the scattering into the umbral shadow much less efficient for near infrared radiation than it is for shorter wavelengths. For all wavelengths, there is a slight asymmetry in the





irradiance gradient at the beginning and at the end of the eclipse. This can be explained similarly to the arguments for radiance in Emde and Mayer (2007), with the contribution function at TOA. Most of the diffuse photons originate from the vicinity of a line (black in Figure 7) at TOA between the point directly above the observer (black dot in Figure 7) and the intersection point of the observer-sun line with the TOA (yellow dot). The histogram in Figure 7 shows the photon density along this line. The

visible asymmetry in this plot is a geometrical effect caused by the deviation of the sun from the zenith position. Because the sun is located in the South-East and the lunar shadow travels from West to East, the bulk in the histogram is covered before the direct beam vanishes at the beginning of totality. As soon as totality starts, irradiance makes a sudden drop. In the second half of totality, the bulk is released step by step, resulting in a measurable increase of irradiance since the direct solar beam is still suppressed. As soon as this is not the case anymore, it outweighs the diffuse radiation.

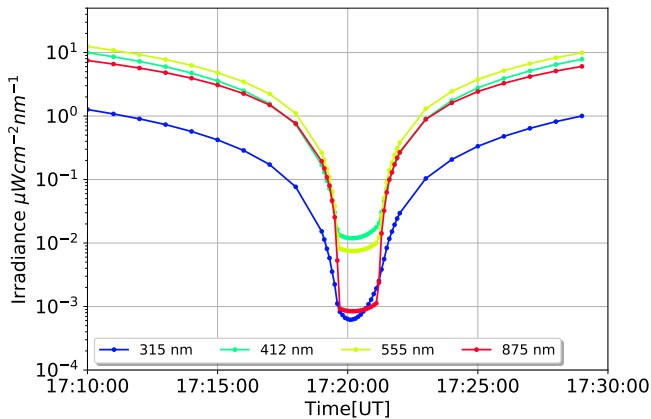

**Figure 6.** Time series of irradiance for different wavelengths from 10 min before to 10 min after totality.

## 3.2    Surface Albedo

So far, the results above were produced with a spatially and spectrally constant Lambertian surface albedo of $0.05$. This does not seem to be a very reasonable assumption since, especially in the near infrared, the albedo over vegetation can reach values up to $0.5$. At the time of the measurements, the environment around the sensor was mainly covered by dried vegetation. The first step to get a more realistic albedo setup is to use a wavelength dependent, spatially constant surface albedo representing

the surrounding area. For this task, the dataset AV87-2 from Clark et al. (1993) was used, which refers to dry, long grass, shown in Figure 8.

     The behavior of irradiance over the full spectrum can be seen in Figure 9. The results are normalized to values obtained with albedo $0.05$ and the same settings otherwise. There is an increase in irradiance of about $20\%$ for the UV wavelengths using the constant surface albedo outside totality (curves for 17:12), which decreases relatively smoothly towards the near infrared.

If however the wavelength dependent dry grass albedo from Figure 8 is applied, there is almost no difference in the irradiance outside totality over the complete spectrum, compared to the albedo $0.05$ surface. The strong increase in the grass' albedo towards longer wavelengths is balanced by the decreased sensitivity of the irradiance in the near infrared towards changes in

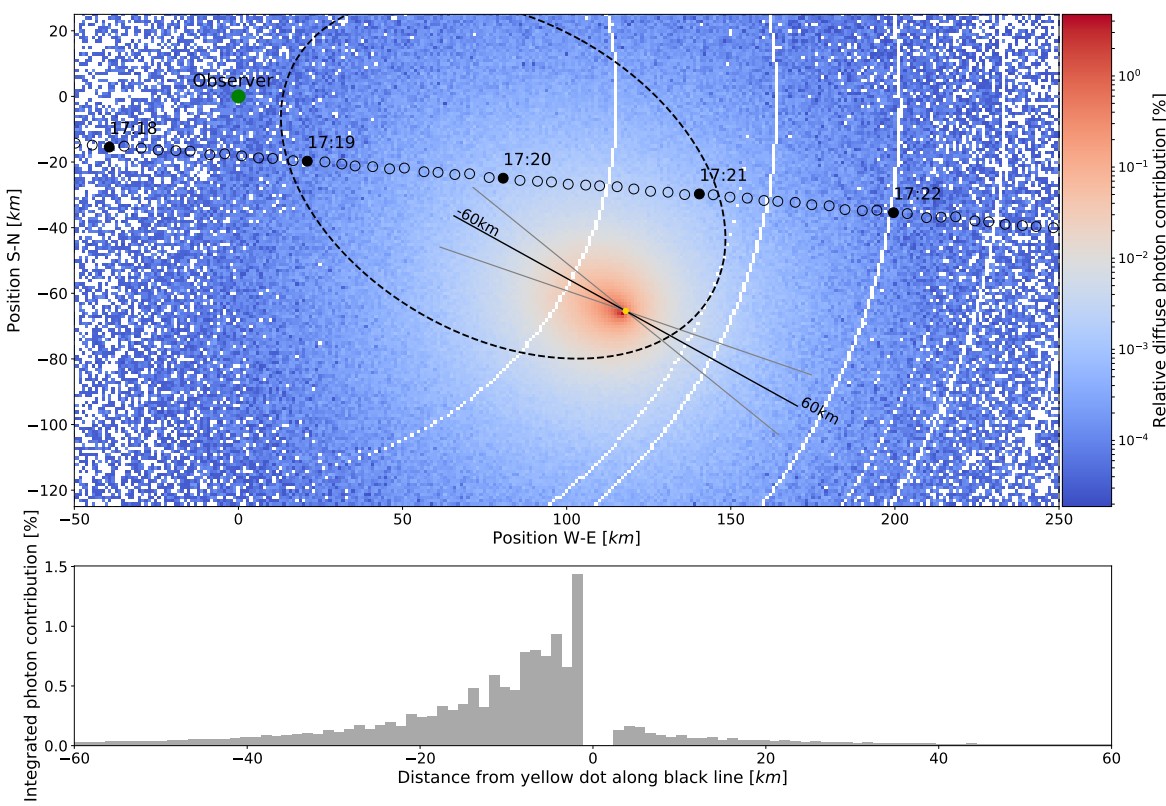

**Figure 7.** The upper image illustrates the relative contribution from each point in the sampling plane to the diffuse irradiance at the observer position (green dot). Shown is an 150 km x 300 km cutout of the domain at 17:20 and 490 nm wavelength. The yellow dot marks the origin of the direct solar radiation. Small black circles indicate the center of the lunar shadow on its way from west to east in 5 s steps with times annotated every minute. The dashed ellipse represents the shape of the umbral shadow at 17:20. The lower histogram shows the contribution function along the black line, integrated within the area bounded by the two grey lines. The white circles visible in the upper image are an artifact from the projection of the spherical grid of the simulation to the flat sampling plane.

the surface reflectance. This is not the case any more as soon as totality has started (curves 17:20). Setting the albedo from $0.05$ to $0.5$, the corresponding curve in Figure 9 reveals a strong wavelength dependence. There are two conspicuous dips around $720$ nm and $940$ nm, indicating a coupling between the albedo sensitivity and water vapor absorption. Absorption and more Rayleigh scattering are also the reason for the lower sensitivity in the ultraviolet compared to the near infrared. In general, an





absorbing medium suppresses longer photons paths, which in turn are more likely to interact with the surface. Qualitatively, this behaviour is also valid for the grass albedo, however with a smaller effect due to the lower albedo values and increasing curves resulting from the grass albedo increase towards longer wavelengths.

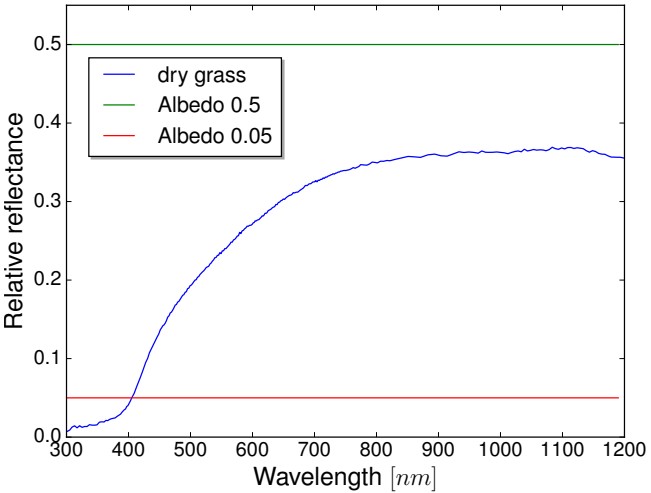

**Figure 8.** Reflectance of dry, brown vegetation, Dataset AV87-2 (Clark et al. (1993)).

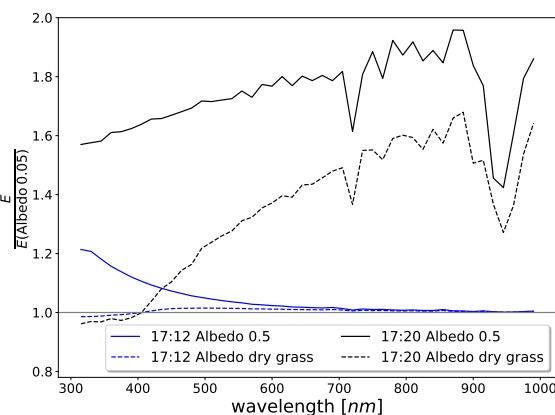

**Figure 9.** Relative spectral irradiance at 17:20 and 17:12 for different spectral, spatially homogenous distributions of the surface albedo. Values are normalized to the corresponding values obtained at 17:12 and 17:20 with an albedo of 0.05.

5      For a future implementation of detailed maps of the surrounding, it is necessary to know the influence of different areas in the simulation domain, in order to choose an appropriate resolution of the maps. Therefore, in this second part of the albedo





study, several idealized simulations were performed with white disks on an otherwise black surface of albedo 0.0, centered around the observer. All following simulations correspond to 17:19:55 and 555 nm wavelength.

In Figure 10, fully filled white disks, centered around the observer, were used. The derivative of the relative irradiance with respect to the radius $r$ of the disks expresses the influence of areas located in a distance $r$. From this, it can be seen that areas

5 in a distance between 20 km and 400 km are the reason for most of the observed changes. The albedo of areas more than 600 km away from the observer has almost no effect on the measurements. In our case, this means the Pacific in the western part of the simulation domain can be neglected. From similar simulations with annuli of different radius, but fixed area, we see that a unit surface in greater distance has less influence on the result. However, this is compensated by the growth of the circle area proportional to $r^2$. This leads to a maximal influence for the region around the observer in 100 km distance. From Figure 10,

10 we conclude that an ideal map should cover the surrounding up to a distance of at least 400 km, possibly at the cost of a lower resolution.

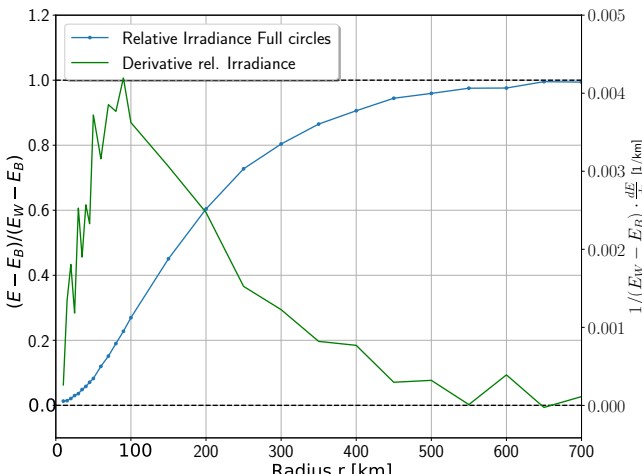

**Figure 10.** Blue graph: change in irradiance at 17:19:55 and 555 nm for white, filled circles with different radii centered around the observer. Shown is the difference to the values from a black surface (subscript $B$), normalized to the difference if the surface is completely white (subscript $W$). In the latter case, irradiance is increased by a factor 2.96 compared to the black surface. The green graph shows the derivative with respect to the circle radius.

### 3.3 Ozone Profile

The total column of ozone (TOC) at the measurement site was derived in Bernhard and Petkov (2019) to be 293 DU directly before the eclipse and 294 DU after the end. These values can be compared with measurements of the TOC from the Ozone

15 Monitoring Instrument (OMI) at the Aura satellite, which measured 298 DU TOC the day before (Veefkind, 2012). In addition, the influence of the vertical distribution of ozone is analyzed using a profile produced with MOZART, the Model for Ozone and Related Chemical Tracers (Emmons et al. (2010)). Compared to the midlatitude summer profile, in the MOZART profile





there are 15 DU ozone shifted from the lower stratosphere between 10 km and 20 km to the altitude range between 20 km and 35 km. Both profiles are scaled to 298 DU TOC.

Results can be seen in Figure 11. There was again no aerosol included in the simulations, but the albedo file for dried grass introduced in subsection 3.2 was used. Outside the totality at 17:12, a decrease in the total ozone column leads to an increase in irradiance like one would expect it for a reduction of an absorbing gas. The sensitivity depends strongly on the ozone cross section, causing changes in irradiance of $\pm 20\%$ at 304 nm and $\pm 0.5\%$ around 600 nm.

A change in the ozone distribution has much smaller influence than a change in the total column outside totality in the region of the Chappuis bands between 400 nm and 650 nm. Here the contribution of the direct irradiance is larger than in the ultraviolet, and direct radiation is not affected by the vertical distribution of the absorber but only by the column. In the UV part of the spectrum, the irradiance increases a little using the profile from MOZART with more ozone in the upper atmosphere. This indicates that most of the photons travel horizontally below 20 km, in accordance with findings in subsection 3.5. Results inside the totality agree with the results outside in the typical features. Due to the missing direct radiation, the sensitivity to ozone is generally increased, with changes of up to $\pm 30\%$ at 304 nm and $\pm 2\%$ at 600 nm. Since the actual intensities are much smaller, the Monte Carlo simulation is more noisy. Without the direct solar radiation, it is possible to see the influence of the difference between MOZART and the midlatitude summer profile around 600 nm as well. The influence is comparable to a change in the TOC of 20 DU. This leads to the conclusion that one has to be careful when deriving the total ozone column during totality from irradiance measurements at the surface.

## 3.4 Topography

Looking at pictures of solar eclipses, one will see a bright, reddish horizon under a black sky, indicating that mountains likely have an influence and should be included to accurately model the data. The site where Bernhard and Petkov (2019) made the measurements was surrounded by a mountain range. The profile given in Table 2 was estimated from photos. To model the effect on the measured irradiance to the first order, we assume completely black mountains which are blocking all the radiance by setting photons coming from angles below $\theta_m$ over the horizon equal to zero. There were three spectral series produced, one without mountains ($\theta_m = 0°$), one with $\theta_m = 10°$ and one with the mountain profile around the sensor at the measurement site 2017 shown in Table 2. The last series will be denoted as *P2017*. The overall effect shares many characteristics with the curves in Figure 9, where the surface albedo was changed. Eight minutes before the totality, changes of up to $2\%$ can be observed for the shorter wavelengths. The increase in direct irradiance is the reason for the vanishing differences at longer wavelengths. Observations during totality show reductions in irradiance of up to $60\%$. This shows that a disproportionally high amount of radiation is coming from the horizon (For isotropically incoming radiance, we would expect only $3\%$ reduction in irradiance.) Like already noticed before, during totality the longer wavelengths are more sensitive towards changes in the surrounding. Again, there are two clearly visible spikes in regions of high absorption, where longer photon paths are suppressed: here, contribution from photons with shorter paths through the atmosphere is enhanced, which are coming from higher angle positions over the hilltops. This agrees with the observation that the sky appears more bluish overhead during an eclipse.





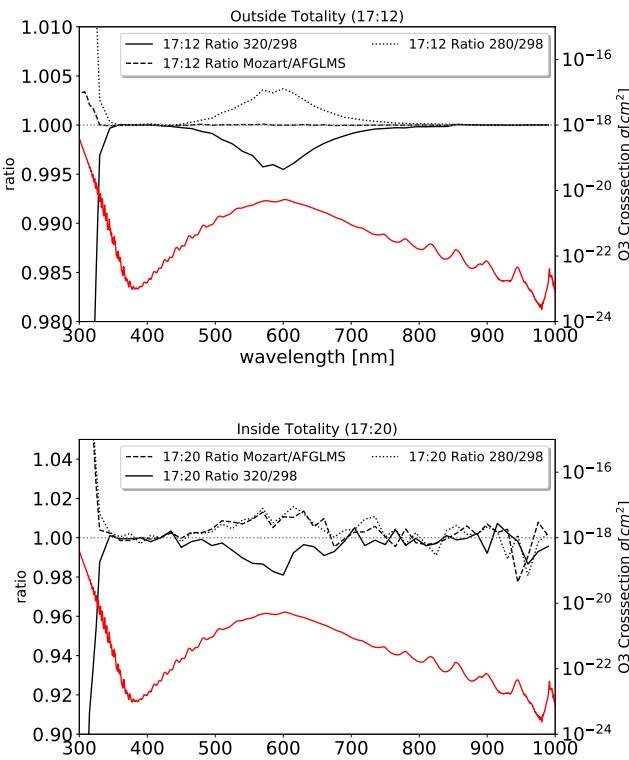

**Figure 11.** Simulated irradiance for different wavelengths, normalized to the irradiance at the corresponding wavelength with the midlatitude summer profile scaled to 298 DU TOC. The solid line corresponds to an increase to 320 DU TOC, the dotted line to a decrease to 280 DU TOC. For the dashed line, the TOC was kept constant to 298 DU TOC, but the midlatitude summer profile was replaced with the one from MOZART. The position of 1.0 is indicated by a coordinate line for better readability. The red curve shows the total ozone absorption cross section, referenced to the right axis. Note the different scales inside and outside totality.

Between the 10° mountains and the P2017 profile, there are no major differences despite the smaller decrease due to the lower average mountain height in P2017.

### 3.5 Photon paths

Many of the effects analyzed in the sections above can be qualitatively understood by looking at the photon paths through the atmosphere. It is important to include the Monte Carlo weighting of the photons, which expresses the effect of the eclipse as well as the air's optical thickness and scattering direction probabilities (For details about the weights, refer to Marshak and Davis (2005)). To simplify the geometry, simulations are done for an observing point directly in the shadow center at 17:20:00 and without any mountains obstructing the horizon. Otherwise, the settings are the same as before, in particular the dry grass albedo from subsection 3.2 and the MOZART ozone profile from subsection 3.3 are used. In Figure 13, the elevation angle



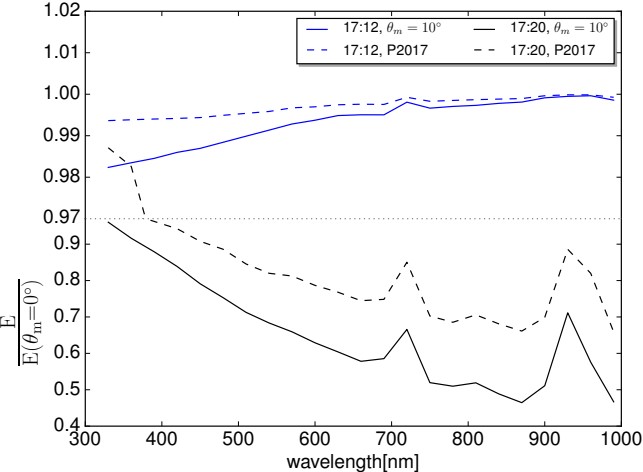

**Figure 12.** Simulated irradiance for different wavelengths and mountain heights, normalized to the irradiance at the corresponding wavelength without mountains ($\theta_m = 0°$). Note the scale change on the y-axis indicated by the dashed, grey line. Blue lines are values outside totality, black lines during totality. Solid lines are for mountains with constant height everywhere around the observer, dashed lines represent the mountain profile "P2017" given in Table 2.

| E | | | | S | | | |
|---|---|---|---|---|---|---|---|
| 0 | 22.5 | 45 | 67.5 | 90 | 112.5 | 135 | 157.5 |
| 7° | 7° | 5° | 5° | 5° | 3° | 2° | 2° |

| W | | | | N | | | |
|---|---|---|---|---|---|---|---|
| 180 | 202.5 | 225 | 247.5 | 270 | 292.5 | 315 | 337.5 |
| 7° | 7° | 3° | 9° | 7° | 7° | 9° | 7° |

**Table 2.** Azimuth angle (second row) and corresponding height of the mountains over the horizon (third row) in °. Values taken from Bernhard and Petkov (2019)

under which the photons arrive at the observer is shown for photons of different scattering order. The results include a cosine weighting to account for the effective area of the sensor. At 306 nm, most of the measured irradiance is coming from photons of order two (20%), contrary to diffuse irradiance under non-eclipse conditions, where single scattered photons make up almost half of the result (not shown). Higher orders than two have monotonically decreasing importance, with 1% contribution from

5    order ten (non-eclipse: 0.2%). The higher order angle distributions approximately follow the $\sin\theta \cdot \cos\theta$ distribution one would expect for isotropically incoming photons, whereas photons of order one show a distortion towards the horizon. For order two and higher, sometimes single photons exhibit an exceptionally high weight, showing up as strong peaks in the histogram. These photons enter the atmosphere with a high initial weight due to the large distance to the umbra (up to 800 km away from the observer) and then travel horizontally 60 km above the surface until they are vertically scattered down to the observer. At 875

10   nm, most of the contribution is coming from doubly scattered photons as well. However, photons with large incident angles are





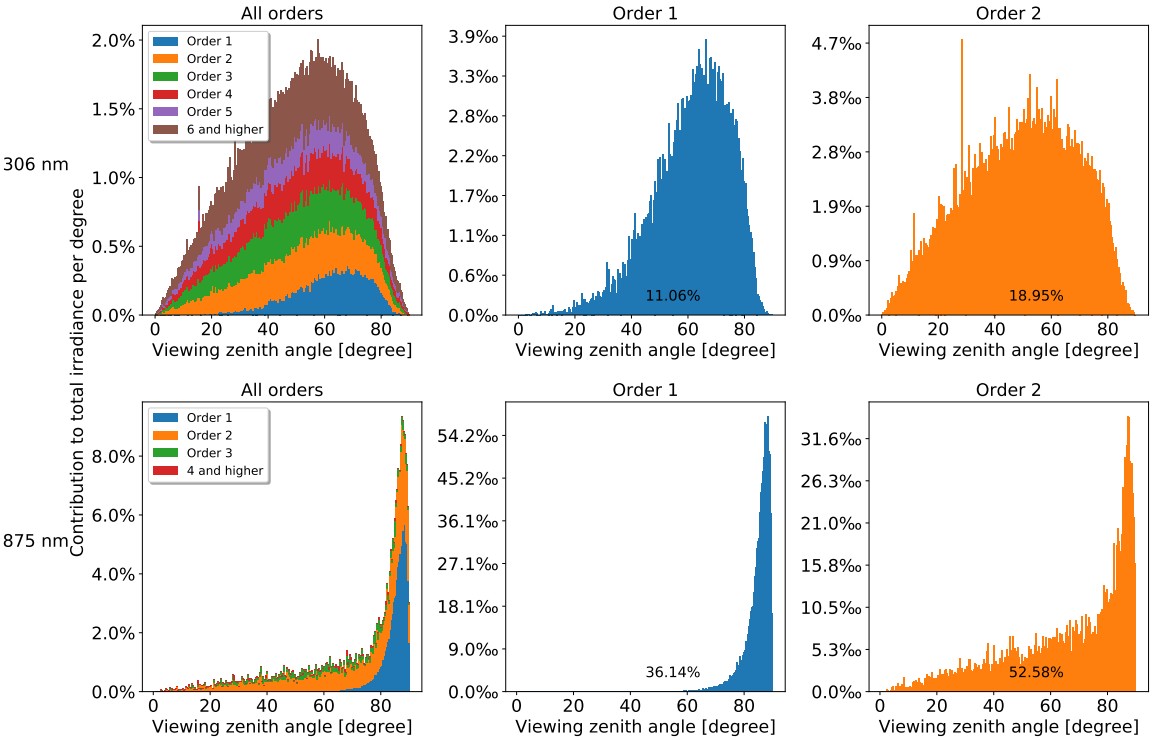

**Figure 13.** Contribution of photons at 306 nm (top) and 875 nm (bottom) to the total spectral irradiance per degree incident angle, split into the contributions from different scattering orders. $0°$ corresponds to zenith direction, $90°$ to the horizon. The left histograms show the stacked combination of the order-specific plots. The black percentages represent the total contribution from photons of this scattering order.

much more important than at 306 nm, as well as for diffuse irradiance under normal conditions at 875 nm. This explains the red horizon during an eclipse and the influence of the topography (subsection 3.4). The underlying reason can be seen in Figure 14, which shows the spatial density of scattering events for photons of order one and two. For the first scattering event, the lunar shadow creates a forbidden volume with the shape of a cylinder in the 3D atmosphere, inclined towards the sun. Longwave

5   photons have their last scattering event on average in a larger distance than shortwave photons (compare Figure 14 a), e) and b), f) ) and therefore larger incident angles. The higher reflectivity of the surface at 875 nm adds to this effect. Actually, most of the multiply scattered photons at longer wavelengths have contact with the Earth's surface, which can be clearly seen in Figure 14 d) and explains the surface influence found in subsection 3.2. Relating the maximum in Figure 14 c) to the one in Figure 14 e), the main photon paths likely contain a $50 - 70$ km horizontal part below 20 km. This is in agreement with the findings in

10   subsection 3.3 that the MOZART profile, with more ozone above 20 km, leads to increased irradiance during totality.







**Figure 14.** Contribution to solar irradiance for photons of different scattering orders and events. 'Order' denotes the total number of scatterings, 'Event' numbers the scattering event on a path from top of atmosphere to the sensor. The horizontal axis denotes the horizontal distance from the observer, the vertical axis the distance from the observer in zenith direction. Contributions are summed radially. Due to the usage of cartesian coordinates in combination with the curvature of the earth, negative scattering heights are possible.

## 3.6 Comparison with measurements

Measurements of global spectral irradiance at different wavelengths were taken by Bernhard and Petkov (2019). Details of the measurement conditions and the calibration process can be found there, the following provides a short summary. The instrument was placed on a field near the Crooked River that flows through Smith Rock State Park and is surrounded by dry gras. In the west and north-east, mountains are obstructing the horizon by up to 12°. The sky this morning was almost clear with only a few clouds near the horizon. It is important to mention that there were several wildfires in the region burning at the measurement time. On the day of the measurement however, the wind direction changed in favor of a decrease of aerosol





loading. The instrument used was a GUVis-3511 multi-channel filter radiometer constructed by Biospherical Instruments Inc. The sensor was placed approximately two meters above the ground. It was equipped with a computer controlled shadowband allowing to calculate direct and global spectral irradiance from the instruments measurements at 306, 315, 320, 340, 380, 395, 412, 443, 490, 532, 555, 665, 875, 940, and 1020 nm with a spectral resolution of 1 nm.

When comparing the measurements around the totality with results from the model, we included the most realistic settings, i.e. we used the spectral dependence of dry grass for the albedo, the ozone profile from MOZART and the mountain profile from Table 2. Furthermore, we specified aerosol in the same manner as proposed by Bernhard and Petkov (2019), based on measurements at 19:00. The column of precipitable water was set to $11\ \mathrm{kgm^{-2}}$ by scaling the midlatitude summer profile. The value was chosen to achieve best agreement between the measurements from 19:00 to 20:00 and the 940 nm channel, which is

strongly influenced by water vapor. This is $20\%$ less than the value of $13.85\ \mathrm{kgm^{-2}}$ given by the ERA-Interim global reanalysis at the place and time of the eclipse (Dee et al., 2011).

    In a first step, these settings are applied to the 1D radiative transfer solver "disort", which is part of libRadtran as well. At 19:00 (20 min after the 4th contact), the results agree with the measurements to better than $5\%$ at each wavelength. This deviation might be caused by uncertainties in the measurements or in the parametrization of the atmospheric state, e.g. the

aerosol loading. To correct the simulation for these errors, which have nothing to do with the eclipse, the simulation values were multiplied by the average deviation between 19:00 and 20:00 for each wavelength.

    The comparison of the results of the 3D calculations and the measurements from 17:18 until 17:23 can be seen in Figure 15, Figure 16 and Figure 17. In each case, the corrected simulations and the measured values are shown in the upper part of the figure. In the lower half, the ratio of observation and simulation is shown. From 17:20:00 until 17:21:15, the shadowband swept

over the sensor, indicated by the grey gap in the time series. In all three domains, the deviations are increased directly before and after the totality, where the temporal gradient of irradiance is very large. Here, the measurements exceed the simulation up to $60\%$ (at 320 nm), giving information about temporal uncertainties which are intensified in these regions. Because the peaks indicate lower simulation results both before and after the totality, the deviation is likely not a temporal shift in one direction, but a difference of 1 - 2 seconds between the actual duration of totality and the period assumed by the simulation. The reasons

could be manifold, e.g. small errors in the lunar disk radius or the moon's deviation from a perfect sphere could change the shadow's size easily and therefore affect the totality time.

    The spectral ratio of measurement and simulation during totality is also shown in Figure 18, together with the ratio of clearsky calculations (i.e. without lunar occultation) to totality simulations. Generally, absorbing wavelengths are attenuated more compared to clearsky conditions, with ranges from 7170 at 350 nm to 140000 at 940 nm. These values agree with the

ones obtained by Bernhard and Petkov (2019).

    Looking at the wavelength domains separately, the best results are achieved between 400 nm and 600 nm (Figure 16). Agreement outside the totality is within $3\%$, during totality the values agree within $10\%$. There are several reasons why these wavelengths are easier to simulate and measure. On the one hand, their clearsky intensities exceed the other domains by at least $50\%$, on the other hand, there are only few absorbing trace gases. However, there seems to be a systematic deviation towards

longer wavelengths, being simulated too high during totality (compare to the downward trend in the black line in Figure 18





between 400 nm and 700 nm). From the sensitivity analysis, one could attribute this observation to the ozone profile, which affects the 500 nm - 700 nm range only during totality. Comparing to Figure 11, a change in $10\%$ for 555 nm might be possible, but would certainly induce changes by more than $10\%$ in the ultraviolet as a consequence.

For 600 nm - 1020 nm (Figure 17), the results are comparable, with somewhat more fluctuations due to the lower solar intensities. The 1020 nm curve shows nearly perfect agreement regarding the intensities, but the temporal uncertainty is one of the highest for all channels. 665 nm shows better temporal agreement, but continues the trend of higher simulation results at totality also described for the 400 nm - 600 nm domain. As already mentioned, the 940 nm channel depends highly on the water vapor concentration in the atmosphere, a quantity with typically large fluctuations in time and space. Nevertheless, the agreement outside the totality is as good as for the other wavelengths in this domain. Of course, one has to remember that the total water vapor column was scaled to fit this channel. Inside the totality, there is too much noise in the 940 nm measurement signal to make a precise statement, although the mean value fits the simulation well.

Between 300 nm and 400 nm (Figure 15), the discrepancy between measurements and simulations is larger compared to the other wavelength domains. Since the intensities are comparably low here, the sensor operates close to its detection limit for 306 nm and 315 nm during totality, which is clearly visible in the large fluctuations. The ratio for 306 nm is not plotted therefore. Outside the totality, all wavelengths are simulated too low. This is likely not attributable to ozone, which affects mostly the wavelengths below 350 nm. It is interesting, that the simulation agrees much better with the measurement at 412 nm than at 395 nm. These wavelengths are close to a critical region in the limb darkening parametrizations, where hydrogen can get fully ionized from the second shell (364.66 nm). Exactly the same behaviour can be seen in Bernhard and Petkov (2019) (compare Figure 9, ibid.) using 1D simulations, even with different limb darkening parametrizations. In addition, 395 nm is right between the strong Ca+ Fraunhofer lines at 393 and 396.5 nm. As a consequence, there is some uncertainty in the measurements from converting the measurements with the GUV's 10 nm wide filters to spectral irradiance at 1 nm resolution. This effect could contribute to the larger difference at 395 compared to 412 nm.

## 4 Conclusions

In this study, the three-dimensional radiative transfer model MYSTIC was used to simulate irradiance at the ground during a total eclipse. The celestial part includes the precise geometry of sun and moon, as well as parametrizations for the limb darkening of the sun. The atmosphere contains realistic gas profiles, a surface with spectrally dependent reflection and a simple topography model. The approach relies only on the time resolved position of sun and moon in space, all further properties like the position, orientation and the elliptical shape of the lunar shadow are derived from this information. The agreement with the observations illustrate the high accuracy of the umbra calculations using the data from JPL Horizons. There are no restrictions on the observer, e.g. it could be out of the shadows center line, like it was the case in this study, or even out of the totality path.

The simulation was applied to the total solar eclipse of August 21, 2017, and the effects of the surface reflectance, ozone profile and the topography were analyzed. In general, we found that the eclipse enhances the sensitivity of global irradiance towards changes in the albedo and topography, which is to a large fraction explained by the absence of direct radiation and the

**Figure 15.** Comparison of simulation and measurements for wavelengths from 300 nm to 400 nm. The upper curves refer to the logarithmic axis on the left and show simulated (solid line) and measured (dotted line) irradiance for different wavelengths. The lower part of each figure shows the measurements divided by the simulation values (dashed line). The x-axis is discontinuous and does not show the period between 17:20:05 and 17:21:05 because the shadowband swiped over the sensor between 17:20:00 and 17:21:10 (grey shading). The vertical, dashed lines show the theoretical start and end of totality.

stronger sensitivity of diffuse irradiance to these parameters. Comparing to diffuse irradiance for near infrared wavelengths under non-eclipse conditions, more radiation is coming from the horizon. The importance of single scattered photons is generally reduced. Higher orders are able to interact with the surface before reaching the observer, which is why the surface reflectance may cause changes up to $200\%$ in the measured irradiance for near infrared wavelengths (completely absorbing versus re-





**Figure 16.** Same as Figure 15, but for wavelengths from 400 nm to 600 nm.

flecting surface). Depending on the wavelength, photons from distances of up to 400 km contribute to the result. Due to the long horizontal photon paths even for ultraviolet wavelengths, not only the total column, but also the vertical distribution of ozone becomes important. Therefore, derivations of the total amount of ozone during totality by comparing surface irradiance measurements at different wavelengths might be misleading. In general, these features of a solar eclipse pose a challenge for

5   models and the assumptions used here are only first order corrections.

The comparison with measurements reveals that solar irradiance over the course of the eclipse can be modeled with high accuracy, considering that solar irradiance drops by four to five orders of magnitude during an eclipse. Showing less than 10%

**Figure 17.** Same as Figure 15, but for wavelengths from 600 nm to 1020 nm.

deviation during the totality at most wavelengths, this also represents a validation of the 3D radiative transfer model MYSTIC as the core of our model.



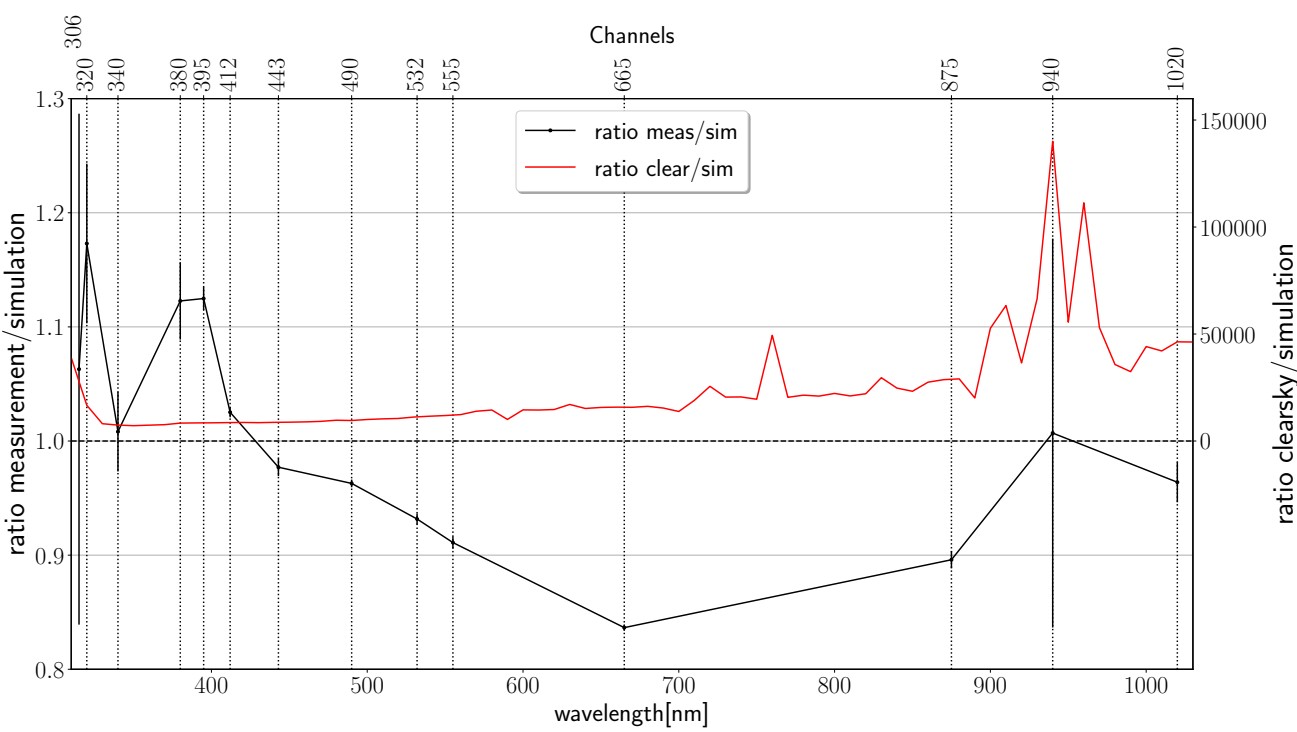

**Figure 18.** Spectral Irradiance during totality (17:20). The black curve shows the deviation of the measurements from the totality simulations, averaged over the 12 data points during totality (17:19:48-17:20:00). The errorbars indicate $1\sigma$-deviation of the original values from the mean. Vertical lines indicate the position of the measurement channels. The red curve shows the ratio of the simulated clearsky irradiance (i.e. without lunar occultation) to simulated totality irradiance at 17:20:00.





*Data availability.* Data available on request.

*Author contributions.* BM and CE developed MYSTIC and the general setup to simulate a solar eclipse. PO further adapted the setup, performed the model analysis and wrote the paper. GB designed and executed the measurements.

*Competing interests.* GB is employed by Biospherical Instruments Inc., which is also the manufacturer of the GUVis-3511 radiometer used for the measurements.





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
