# Peer review of "Accurate 3D radiative transfer simulation of spectral solar irradiance during the total solar eclipse of August 21, 2017"

_Atmospheric Chemistry and Physics, 2019_

## Referee Comment (RC1) · Anonymous Referee #2 · 12 Sep 2019

**General comments**

This paper presents 3D radiative transfer simulations of surface spectral irradiance near and in the umbra shadow during 21 August 2017 eclipse. The study provided detailed sensitivity analysis based on a well-recognized 3D radiative transfer model (MYSTIC). They showed that the irradiance is sensitive to spectral albedo, ozone vertical distribution, and surrounding mountains. They found that simulations agree reasonably well with observations. The results are very useful for the community for understanding the radiative transfer processes during a solar eclipse. The paper is well-organized and addresses relevant scientific questions within the scope of ACP. I recommend it for

publication in ACP after a minor revision.

My main concern is that the simulations are under clear atmospheric conditions without discussion about possible cloud effects. Bernhard and Petkov (2019) mentioned that "The sky was free of clouds in the direction of the Interactive Sun, with small clouds lingering only near th horizon." This is consistent with comment GOES-16 images (see http://www2.mmm.ucar.edu/imagearchive/ and http://rammbslider.cira.colostate.edu/?sat=goes-16&z=4&im=12&ts=1&st=20170821174538&et=20170821200037&speed=130&motio 1&hide controls=0&mouse draw=0&follow feature=0&follow hide=0&s=rammbslider&sec=full disk&p%5B0%5D=geocolor&x=9288&y=3872). Even though thin cirrus clouds were not in the direction of the observer and the Sun, they could be potentially important generating significant amount of diffuse radiation reaching the surface near and in the totality region. Could the presence of thin cirrus clouds explain the larger observed spectral irradiance compared to the model simulations? Maybe, a simple simulation for a fraction of thin cirrus cloud not shading the Sun could help to answer this question. The results could be included in discussion section.

Some specifics:

1. Spell out MYSTIC in line 7 in abstract and line 54 in introduction section.

2. "contribution function" is introduced in line 70 of section 2. This is an important concept in this paper. However it is not very clear to me. A better explanation will be helpful.

3. Remove Mayer (2009); Emde and Mayer (2007) in line 50 in section 2.2 since they were already referenced before.

4. Is w(d0,lambda,t) e-tau in line 16 of section 2.3 the direct component for partial eclipse conditions? If so, make it clear.

5. "Because its time dependence is in the order of weeks, it is kept constant during the eclipse." in line 25 of section 2.3. Consider to remove this sentence since it was
referenced.

6. "The spectral irradiance of the 315 nm wavelength is remarkably lower than the visible irradiances, which is mainly due to ozone absorption." Is that really true? Perhaps one needs to compare irradiance during the totality with that in partial eclipse. If we look at 17:15:00, the relative difference between irradiance at 315 nm and 555 nm is about that same. I understand that ozone absorption reduces the irradiance at 315 nm reaching the troposphere, where most of the scattering takes place. Of course, the Rayleigh scattering is strongest among all wavelength considered. A better discussion will help readers to understand this point.

7. Figure 7 is very confusing. "Most of the diffuse photons originate from the vicinity of a line (black in Figure 7) at TOA between the point directly above the observer (black dot in Figure 7) and the intersection point of the observer-sun line with the TOA (yellow dot)." in line 65 of section 3.1. Not sure about a line at TOA because the scattering occurs in the atmosphere. "Black dot" should be green dot? "the vicinity of a line" is confusing. It seems that most of the diffuse photons originate from an area about 20 km in radius around the yellow dot. Is figure 7 (upper panel) for the partial eclipse at the observer's site since it says for 17:20 and the observer is out of the ellipse? If possible, it would be nice to have the artifact white circle removed.

8. In Figure 9 caption, suggest to indicate 17:20 (totality) and 17:12 (partial eclipse).

9. A discussion of the physical mechanism of different surface albedo on surface irradiance for partial and total eclipse will be useful (first paragraph on pager 6).

10. Figure 10 is hard to understand. Why "From this, it can be seen that areas in a distance between 20 km and 400 km are the reason for most of the observed changes." (line 43 on page 6)? The purpose of this is to assess the validity of assumption of spatially uniform albedo, which may not valid for the Pacific Ocean. Perhaps ignore this figure and just make a statement to about the result: surface albedo is not important beyond 400 km, and the results of simulations will not change even if the albedo is
different from the ocean albedo for the Pacific.

11. line 80 on page 6, "+-20% at 304 nm". 20% is out of scale in Figure 11. At least mention it in the figure caption.

12. Paragraph line 81 on page 6, "Here the contribution of the direct irradiance is larger than in the ultraviolet, and direct radiation is not affected by the vertical distribution of the absorber but only by the column." Perhaps mention ozone absorption optical depth at the peak of the Chappuis bands is about 0.04 compared to the absorption optical depth of 1.8 at 304 nm, or a sentence about ozone absorption cross section in Fig. 11 since it is already there.

13. Would be nice to have the same legends for both upper and lower panels in Figure 11.

14. line 4 on page 7, "+-30% at 304 nm". But 30% is out of scale in Figure 11.

15. line 3 on page 8, "The last series will be denoted as P2017. The overall effect shares many characteristics with the curves in Figure 9, where the surface albedo was changed. Eight minutes before the totality, changes of up to 2% can be observed for the shorter wavelengths. The increase in direct irradiance is the reason for the vanishing differences at longer wavelengths. Observations during totality show reductions in irradiance of up to 60%. This shows that a disproportionally high amount of radiation is coming from the horizon (For isotropically incoming radiance, we would expect only 3% reduction in irradiance.)..." I don't get it. Figure 9 shows the theoretical calculations. I don't see why changes of irradiance (relative to what?) can be observed.

16. Description of Figure 12 should be included in the text. Maybe in the last paragraph of section 4.2.

17. Line 45 on page 10, "To correct the simulation for these errors, which have nothing to do with the eclipse, the simulation values were multiplied by the average deviation between 19:00 and 20:00 for each wavelength." Do not understand this? Also, do you

ACPD
assume that the atmospheric state didn't change?

18. Suggest to combine figures 15, 16, 17 to Figure 15a, 15b, 15c to reduce the number of figures. Suggest to remove "ratio" since the "ratio" is already in the title of Y-axis.

---

## Author Comment (AC1) · 19 Sep 2019

**Answers to Referee 1**

September 17, 2019

**Influence of clouds**

**Comment by Referee**

My main concern is that the simulations are under clear atmospheric conditions without discussion about possible cloud effects. Bernhard and Petkov(2019) mentioned that "The sky was free of clouds in the direction of the Sun, with small clouds lingering only near the horizon." This is consistent with GOES-16 images (see http://www2.mmm.ucar.edu [...] Even though thin cirrus clouds were not in the direction of the observer and the Sun, they could be potentially important generating significant amount of diffuse radiation reaching the surface near and in the totality region. Could the presence of thin cirrus clouds explain the larger observed spectral irradiance compared to the model simulations? Maybe, a simple simulation for a fraction of thin cirrus cloud not shading the Sun could help to answer this question. The results could be included in discussion section.

**Authors' Response**

The amount of clouds were really negligible. The sentence mentioned in Bernhard and Petkov (2019) refers to the clouds visible in Figure 1. Besides these, the only other feature visible on the clear sky is a thin layer close to the horizon in the gap between the mountains in the far right in Figure 2. It is not clear, whether they are attributable to clouds or aerosols. Looking at satellite pictures, the larger area around the observation spot was mostly free of clouds as well (Compare to Figure 3 and Figure 4). Concerning these circumstances, clear atmospheric conditions seemed the most reasonable assumption for our setup.

**1 Comment**

**1.1 Comment by Referee**

Spell out MYSTIC in line 7 in abstract and line 54 in introduction section.

**1.2 Authors' Response**

Changed accordingly.

**1.3 Changes to manuscript**

...the radiative transfer model MYSTIC (Monte Carlo code for the phYSically correct Tracing of photons In Cloudy atmospheres;Mayer (2009); ...

**2 Comment**

**2.1 Comment by Referee**

"contribution function" is introduced in line 70 of section 2. This is an important concept in this paper. However it is not very clear to me. A better explanation will be helpful.

**2.2 Authors' Response**

We tried to make the concept more clear in section 2.2., especially by adding the following explanation.

**2.3 Changes to manuscript**

If a photon leaves at the TOA into the direction of the sun, it is traced further to the Sampling Plane (compare Figure 1), where it is counted at the incident position. Averaging over many photons, this creates a map of contributions, i.e. how much every point in the sampling plane contributes to the irradiance at the observer position. We call this map the contribution function $C(\vec{d}, \lambda, t)$.

**3 Comment**

**3.1 Comment by Referee**

Remove Mayer (2009); Emde and Mayer (2007) in line 50 in section 2.2 since they were already referenced before.

**3.2 Authors' Response**

Changed accordingly

**4   Comment**

**4.1   Comment by Referee**

Is w(d0,lambda,t) e-tau in line 16 of section 2.3 the direct component for partial eclipse conditions? If so, make it clear

**4.2   Authors' Response**

Yes, it is the direct component under partial conditions, if evaluated at the corresponding times. $w$ describes the intensity of solar irradiance at every position and time in the sampling plane, including the moon's shadow. Evaluating it at the described position $\vec{d_0}$ and multiplying by the atmospheric transmittance, we get direct radiation under non-, partial or total eclipse conditions (=0 in the latter case).

**4.3   Changes to manuscript**

Because for partial eclipse conditions, the light of the sun reaching the sensor directly...

**5   Comment**

**5.1   Comment by Referee**

"Because its time dependence is in the order of weeks, it is kept constant during the eclipse." in line 25 of section 2.3. Consider to remove this sentence since it was referenced.

**5.2   Authors' Response**

Sentence was removed and replaced by: The factor $\kappa(t)$ corrects the ETS for variations in the earth-sun distance by use of the formula from Iqbal1983, evaluated for August 21.

**6   Comment**

**6.1   Comment by Referee**

"The spectral irradiance of the 315 nm wavelength is remarkably lower than the visible irradiances, which is mainly due to ozone absorption." Is that really true? Perhaps one needs to compare irradiance during the totality with that in partial eclipse. If we look at 17:15:00, the relative difference between irradiance at 315 nm and 555 nm is about that same. I understand that ozone absorption reduces the irradiance at 315 nm reaching the troposphere, where most of the scattering takes place. Of course, the Rayleigh scattering is strongest among all

wavelength considered. A better discussion will help readers to understand this point.

**6.2 Authors' Response**

We changed the sentence in the following way:

**6.3 Changes to manuscript**

The spectral irradiance of the 315 nm wavelength is remarkably lower than the visible irradiances, which is mainly due to the naturally lower irradiance values under non-eclipse condition.

**7 Comment**

**7.1 Comment by Referee**

Figure 7 is very confusing. "Most of the diffuse photons originate from the vicinity of a line (black in Figure 7) at TOA between the point directly above the observer (black dot in Figure 7) and the intersection point of the observer-sun line with the TOA (yellow dot)." in line 65 of section 3.1. Not sure about a line at TOA because the scattering occurs in the atmosphere. "Black dot" should be green dot? "the vicinity of a line" is confusing. It seems that most of the diffuse photons originate from an area about 20 km in radius around the yellow dot. Is figure 7 (upper panel) for the partial eclipse at the observer's site since it says for 17:20 and the observer is out of the ellipse? If possible, it would be nice to have the artifact white circle removed.

**7.2 Authors' Response**

Indeed, this figure might be difficult to understand at the moment. The key point is, that the contribution function in the sampling plane is shown, located 120km above the observer. To enhance understanding, we propose to change Figure 1 in the manuscript according to Figure 5, which should explain the whole setup better. In Figure 5, we introduced the location of the green and yellow points as defined in Figure 7 of the manuscript. Basically, Figure 5 provides a side view of the setup, while Figure 7 in the manuscript can be interpreted as a view from above.

The "black dot" in the manuscript should of course be "green dot".

Concerning the white circles: In our setup, they are difficult to remove technically. Because they are present in the contribution function in the model, we decided to show them in Figure 7 in the manuscript as well.

**7.3 Changes to manuscript**

Caption of Figure 7 in manuscript: "The upper image illustrates the relative contribution from each point in the sampling plane to the diffuse irradiance. Shown is a 150 km x 300 km cutout of the contribution function at 17:20 and 490 nm wavelength. The green dot is located directly above the observer, the yellow dot marks the origin of the direct solar radiation (compare to Figure 1). Small black circles indicate the center of the lunar shadow on its way from west to east in 5 s steps with times annotated every minute. The dashed ellipse represents the shape of the umbral shadow at 17:20. The lower histogram shows the contribution function along the black line, integrated within the area bounded by the two grey lines. The white circles visible in the upper image are an artifact from the projection of the spherical grid of the simulation to the flat sampling plane."

**8 Comment**

**8.1 Comment by Referee**

In Figure 9 caption, suggest to indicate 17:20 (totality) and 17:12 (partial eclipse).

**8.2 Authors' Response**

Changed accordingly.

**9 Comment**

**9.1 Comment by Referee**

A discussion of the physical mechanism of different surface albedo on surface irradiance for partial and total eclipse will be useful (first paragraph on pager 6)

**9.2 Authors' Response**

Please specify, which features should be explained in more detail, because some of them are already discussed in the paragraph. We added the following sentence:

**9.3 Changes to manuscript**

[...which decreases relatively smoothly towards the near infrared.] This is due to the smaller contribution of diffuse irradiance, which is the only part affected by changes in the albedo, in the near infrared under partial eclipse conditions. [If however...]

**10  Comment**

**10.1  Comment by Referee**

Figure 10 is hard to understand. Why "From this, it can be seen that areas in a distance between 20 km and 400 km are the reason for most of the observed changes." (line 43 on page 6)? The purpose of this is to assess the validity of assumption of spatially uniform albedo, which may not valid for the Pacific Ocean. Perhaps ignore this figure and just make a statement to about the result: surface albedo is not important beyond 400 km, and the results of simulations will not change even if the albedo is different from the ocean albedo for the Pacific.

**10.2  Authors' Response**

So far, we have changed the surface albedo globally, without discussing the importance of different areas. With Figure 10, we wanted to make a quantitative statement about the importance of surface areas within a certain distance. "20km-400km" comes from the fact, that the jump from 0 to 1 of the blue curve occurs almost completely between 20km and 400km (90%). Should a future study decide to implement a detailed map of the surface, we hope to provide some advice on the necessary coverage and resolution with these results.

**11  Comment**

**11.1  Comment by Referee**

11. line 80 on page 6, "+-20% at 304 nm". 20% is out of scale in Figure 11. At least mention it in the figure caption.

**11.2  Authors' Response**

Changed accordingly.

**12  Comment**

**12.1  Comment by Referee**

Paragraph line 81 on page 6, "Here the contribution of the direct irradiance is larger than in the ultraviolet, and direct radiation is not affected by the vertical distribution of the absorber but only by the column." Perhaps mention ozone absorption optical depth at the peak of the Chappuis bands is about 0.04 compared to the absorption optical depth of 1.8 at 304 nm, or a sentence about ozone absorption cross section in Fig. 11 since it is already there.

**12.2 Authors' Response**

Changed accordingly.

**12.3 Changes to manuscript**

[...changes in irradiance of $\pm 20\%$ at 304 nm and $\pm 0.5\%$ around 600 nm] (At 304 nm, the optical depth due to ozone is around 2.1, in the Chappuis bands between 400 nm and 650 nm, it peaks at 0.05).

**13 Comment**

**13.1 Comment by Referee**

Would be nice to have the same legends for both upper and lower panels in Figure 11.

**13.2 Authors' Response**

We changed the Figure according to Figure 6.

**14 Comment**

**14.1 Comment by Referee**

line 4 on page 7, "+-30% at 304 nm". But 30% is out of scale in Figure 11.

**14.2 Authors' Response**

See section 11

**15 Comment**

**15.1 Comment by Referee**

line 3 on page 8, "The last series will be denoted as P2017. The overall effect shares many characteristics with the curves in Figure 9, where the surface albedo was changed. Eight minutes before the totality, changes of up to 2% can be observed for the shorter wavelengths. The increase in direct irradiance is the reason for the vanishing differences at longer wavelengths. Observations during totality show reductions in irradiance of up to 60%. This shows that a disproportionally high amount of radiation is coming from the horizon (For isotropically incoming radiance, we would expect only 3% reduction in irradiance.). . ." I don't get it. Figure 9 shows the theoretical calculations. I don't see why changes of irradiance (relative to what?) can be observed.

**15.2  Authors' Response**

There was definitely a reference missing in the manuscript. The paragraph refers primarily to Figure 12.

**15.3  Changes to manuscript**

The last series will be denoted as *P2017*. The results can be seen in Figure 12.

**16  Comment**

**16.1  Comment by Referee**

Description of Figure 12 should be included in the text. Maybe in the last paragraph of section 4.2.

**16.2  Authors' Response**

See section 15

**17  Comment**

**17.1  Comment by Referee**

Line 45 on page 10, "To correct the simulation for these errors, which have nothing to do with the eclipse, the simulation values were multiplied by the average deviation between 19:00 and 20:00 for each wavelength." Do not understand this? Also, do you assume that the atmospheric state didn't change?

**17.2  Authors' Response**

Yes, our model does not include any changes in the atmosphere, especially none that might be caused by the eclipse itself (like for example an eclipse feedback on the amount of ozone, see e.g. Zerefos et. al. 2000). All of our parameterization are either general standard values (the midlatitude summer profile), or derived from irradiance measurements after the eclipse at 19.00 UTC like water vapor concentration or the aerosol optical depth. With these values, we can model non-eclipse irradiances with less than 5% deviation to the measurements around 19.00. To correct for these non eclipse-related deviations, all simulated channels were multiplied by a correction factor to fit the time from 19.00 to 20.00 (the procedure is similar to Bernhard & Petkov 2019, section 7.1).

**18 Comment**

**18.1 Comment by Referee**

Suggest to combine figures 15, 16, 17 to Figure 15a, 15b, 15c to reduce the number of figures. Suggest to remove "ratio" since the "ratio" is already in the title of Y-axis

**18.2 Authors' Response**

Changed accordingly.

[Figure]

Figure 1: Picture taken at 9:57 local time (partial eclipse already started at 9.06), about 23 minutes before the start of totality.

[Figure]

Figure 2: Picture taken at 9:25 local time.

[Figure]

Figure 3: The large scale situation around the measurement site (red dot) at August 21, 2017. The red circle shows a 175 km radius. The white plumes visible in this area are from wildfires. Corrected Reflectance MODIS Image from Terra satellite, accesible via NasaWorldView: `https://worldview.earthdata.nasa.gov/?v=-125.74822628736678,42.11048430043591,-117.31072628736678,46.68958586293591&t=2017-08-21-T06%3A00%3A00Z&l=Reference_Labels,Reference_Features(hidden),Coastlines,VIIRS_SNPP_CorrectedReflectance_TrueColor(hidden),MODIS_Aqua_CorrectedReflectance_TrueColor(hidden),MODIS_Terra_CorrectedReflectance_TrueColor`

[Figure]

Figure 4: Picture taken by Goes 16 imager. The shadow of the moon is visible in the eastern part of the US. Snapshot from video available via `https://www.goes-r.gov/multimedia/originalVideoCopies/dataAndImagery/GOES16/conus_leaving_east_coast.mp4`

[Figure]

Figure 5: Schematic illustration of a solar eclipse, including umbral (dark gray) and penumbral (light gray) shadow and the definition of sampling plane and TOA (Top Of Atmosphere). S denotes the sun, M the moon and O the observer. The green dot is located vertically above the observer. The yellow dot represents the intersection of the observer-sun line with the sampling plane. The dashed lines show exemplary photon paths. In the atmospheric part, the solar angle is assumed to be constant, atmospheric refraction is neglected.

[Figure]

[Figure]

Figure 6: Simulated irradiance for a partial eclipse (panel (a)) and totality (panel (b)) for different wavelengths, normalized to the irradiance at the corresponding wavelength with the midlatitude summer profile scaled to 298 DU TOC. The solid line corresponds to an increase to 320 DU TOC, the dotted line to a decrease to 280 DU TOC. For the dashed line, the TOC was kept constant to 298 DU TOC, but the midlatitude summer profile was replaced with the one from MOZART. The position of 1.0 is indicated by a coordinate line for better readability. The red curve shows the total ozone absorption cross section, referenced to the right axis. For $304nm$, changes are $\pm20\%$ outside totality and $\pm30\%$ inside (not shown).

---

## Referee Comment (RC2) · Anonymous Referee #3 · 25 Nov 2019

The authors of this paper are trying to simulate the radiative transfer conditions under a total solar eclipse. To this direction they used the 3D model MYSTIC providing analytical explanation of almost all geometries and physical effects. The methodologies used are scientifically correct and the accuracy of simulations was precisely defined taking into account the majority of atmospheric parameters that affect solar irradiance.

However some assumptions made in the whole approach drive me to recommend the manuscript for publication in the Atmospheric Chemistry and Physics journal after minor correction. These assumptions have to do with the exclusion from the simulations of the aerosol and cloud effects.

[Figure]

Since the first word of the title is "Accurate", I totally agree that it is important to analyze e.g. the different impact of albedo dry grass vs spectrally steady albedo values or the impact of various mountain heights and the ozone profile. But at the same time it is at least equally important to adequately handle (page 8, lines 8-9; page 13, lines 3-6; page 18, lines 5-16) the aerosol presence and loading (measurements used just at 19:00). Simultaneously, based on images from satellites the potential impact from the wildfires and the presence of clouds (few but important for accurate 3D simulations) has to be specified and quantified in terms of contribution to the overall impact, and MYSTIC is ideal for cloudy atmospheres.

In particular, based on the satellite images there are both clouds and aerosols present in the atmosphere before, during and after the solar eclipse, but there was not performed a relevant simulation taking into account their potential contribution into e.g. the diffuse radiation. I suggest the authors to include some sensitivity analysis taking into account the microphysics and optical characteristics of the existing formations (possibly cirrus) and loads (biomass burning aerosols). It makes no sense to show 18 figures (some of them are difficult to understand) and not to provide a figure for the potential effect of clouds and aerosols.

Overall this is a valuable study for the radiative transfer modeling community and the approach and analysis followed is complete. As a result I believe that merits publication in the Atmospheric Chemistry and Physics journal taking first into account the above suggestions. The proposed revision will help in the direction of reaching the holism of the authors analytical thinking and for the paper to be one of the few that indeed provide such accurate and 3D radiative transfer simulations during a total solar eclipse.
* * *

---

## Author Comment (AC2) · 29 Dec 2019

**Answers to Referee 3**

December 23, 2019

**1 Influence of aerosol**

**1.1 Comment by Referee**

[...]But at the same time it is at least equally important to adequately handle (page 8, lines 8-9; page 13, lines 3-6; page 18, lines 5-16) the aerosol presence and loading (measurements used just at 19:00).[...]

**1.2 Authors' Response**

We agree with Referee #3 that aerosols are another important factor. So far, aerosols were included in the final simulations, but were not discussed in the sensitivity analysis, mainly to keep the study short and because they were already discussed in Emde et al. 2007, however only for wavelengths between 300 nm and 500 nm. Following the Referee's suggestion, we include the following section together with an additional Figure in the manuscript:

**1.3 Changes in manuscript**

Aerosols

Another factor which influences direct as well as diffuse irradiance is aerosol. During the measurements by Bernhard et al. 2019, used for the comparison in subsection 3.6, several wildfires were burning in the region. Therefore, the authors derived aerosol optical depth (AOD) from measurements of direct spectral irradiance for each channel. At times 16:03 and 19:00, i.e. before the first and after the fourth contact, they fitted Ångström functions of the form $\tau = \beta \lambda^{-\alpha}$ to the data. The coefficients for the first measurement are $\alpha = 1.96$ and $\beta = 0.057$, for the second measurement they got $\alpha = 2.1$ and $\beta = 0.0394$. In our reference simulation, we specify aerosol the same way as they did and similar to Emde et al. 2007. This includes an aerosol model of Shettle 1990. Single scattering albedo was set to 0.95 and asymmetry parameter to 0.7. This profile is scaled to an AOD described by the Ångström function at 19:00. Furthermore, the settings from the sections above are applied: dry grass albedo, MOZART ozone profile and realistic mountain profile. In Figure 1, the blue curve shows the relative change in irradiance during totality if no aerosol would be present. In the

near infrared the effect is strongest, with signal reductions up to 60% despite the generally lower AOD. To get an estimate of the uncertainty of the reference simulation, the orange curve was produced with the Ångström parameters from 16:03. Maximal changes are 10%. To summarize, consideration of aerosol is very important, however the effect of aerosol differences before and after the eclipse were smaller than 10%. Again, we see the red and near infrared wavelength to be more sensitive to the surrounding during totality, which we will explain in the following section.

Abstract:

[...]The influence of the surface reflectance, the ozone profile, mountains surrounding the observer and aerosol is investigated. An increased sensitivity during totality is found for the reflectance, aerosol and topography, compared to non-eclipse conditions.[...]

Introduction:

[..]as well as the surface reflectance, topography and aerosol[..]

[Figure]

Figure 1: Simulated irradiance during totality (17:20) for different wavelengths, relative to irradiance with the Ångström parameters derived at 19:00. For the blue curve, no aerosol was specified, for the orange curve aerosol was parameterized with the Ångström parameters derived at 16:03.

**2  Influence of clouds**

**2.1  Comment by Referee**

I suggest the authors to include some sensitivity analysis taking into account the microphysics and optical characteristics of the existing formations (possibly cirrus)

**2.2  Author's Response**

Clouds are definitely an interesting factor, especially because they are likely present in most eclipse observations. However, as we already discussed with Referee #2, in our case the visible sky as well as the larger area were nearly

cloud free. Therefore, for modelling the observations, a cloudless setup seemed most reasonable to us. A general sensitivity analysis would be interesting as well, but would require non-trivial changes to our model which does not permit simulations with three-dimensional clouds, spherical geometry, and eclipse condition. These are planned as a future project.